# Cytoplasmic Localization of Thyroid Hormone Receptor (TR) Alpha and Nuclear Expression of Its Isoform TRα2 Determine Survival in Breast Cancer in Opposite Ways

**DOI:** 10.3390/cancers15143610

**Published:** 2023-07-13

**Authors:** Mariella Schneider, Melitta B. Köpke, Alaleh Zati zehni, Theresa Vilsmaier, Mirjana Kessler, Magdalena Kailuweit, Aurelia Vattai, Helene Hildegard Heidegger, Vincent Cavaillès, Udo Jeschke, Nina Ditsch

**Affiliations:** 1Department of Obstetrics and Gynecology, University Hospital Augsburg, 86156 Augsburg, Germany; mariella.schneider@uk-augsburg.de (M.S.); melitta.koepke@uk-augsburg.de (M.B.K.); nina.ditsch@uk-augsburg.de (N.D.); 2Department of Obstetrics and Gynecology, University Hospital Munich, LMU Munich, 81377 Munich, Germanytheresa.vilsmaier@med.uni-muenchen.de (T.V.); mirjana.kessler@med.uni-muenchen.de (M.K.); magdalena.kailuweit@swmbrk.de (M.K.); aurelia.vattai@med.uni-muenchen.de (A.V.); helene.heidegger@med.uni-muenchen.de (H.H.H.); 3IRCM—Institut de Recherche en Cancérologie de Montpellier, INSERM U1194, Université Montpellier, Parc Euromédecine, 208 rue des Apothicaires, CEDEX 5, F-34298 Montpellier, France; vincent.cavailles@inserm.fr

**Keywords:** breast cancer, thyroid hormone receptor, TRα, TRα1, TRα2, subcellular localization, prognosis, overall survival, disease-free survival, immuno-histochemistry

## Abstract

**Simple Summary:**

There is evidence of a link between breast cancer and thyroid disease. Patients with thyroid dysfunction have an increased incidence of breast cancer compared to healthy women. Therefore, the aim of this study was to evaluate the relevant prognostic value of nuclear and cytoplasmic thyroid receptor (TR) alpha expression and its α1 and α2 isoforms in breast cancer. TRα expression was found to play a contradictory role in BC prognosis depending on its intracellular localization: our results show that TRα and TRα2 expression play different prognostic roles depending on their subcellular localization. Cytoplasmic TRα was a negative prognosticator, whereas nuclear TRα2 expression was positively associated with overall survival. This study highlights the need to further investigate the behavior of TR depending on their intracellular localization. The significance of their subcellular expression and interaction with other members of the nuclear receptor family needs to be elucidated to find new treatment options for breast cancer in the future.

**Abstract:**

The aim of this retrospective study was to assess the respective prognostic values of cytoplasmic and nuclear TRα, TRα1, and TRα2 expression in breast cancer (BC) tissue samples and correlate the results with clinico-pathological parameters. In 249 BC patients, the expression patterns of general TRα and the α1 and α2 isoforms were evaluated via immuno-histochemistry. Prognosis-determining aspects were calculated via univariate, as well as multivariate, analysis. Univariate Cox-regression analysis revealed no association between nuclear TRα expression and overall survival (OS) (*p* = 0.126), whereas cytoplasmic TRα expression was significantly correlated with a poor outcome for both OS (*p* = 0.034) and ten-year survival (*p* = 0.009). Strengthening these results, cytoplasmic TRα was found to be an independent marker of OS (*p* = 0.010) when adjusted to fit clinico-pathological parameters. Analyses of the TRα-subgroups revealed that TRα1 had no prognostic relevance, whereas nuclear TRα2 expression was positively associated with OS (*p* = 0.014), ten-year survival (*p* = 0.029), and DFS (*p* = 0.043). Additionally, nuclear TRα2 expression was found to be an independent positive prognosticator (*p* = 0.030) when adjusted to fit clinico-pathological parameters. Overall, our results support the hypothesis that subcellular localization of TRα and its isoforms plays an important role in the carcinogenesis and prognosis of breast cancer. Cytoplasmic TRα expression correlates with more aggressive disease progression, whereas nuclear TRα2 expression appears to be a protective factor. These data may help us to prioritize high-risk BC subgroups for possible targeted tumor therapy.

## 1. Introduction

Breast cancer (BC) is the most common cancer and leading cause of cancer death worldwide [1,2]. According to the World Health Organization (WHO), 2.3 million women were diagnosed with BC in 2020, and 685,000 deaths were BC-related [3]. As a highly heterogeneous disease, BC diagnosis and treatment are complex and differ according to clinical tumor subtypes [4,5]. Opportunities for breast cancer therapies have evolved tremendously in recent decades, offering a variety of therapeutic approaches depending on whether the therapy required is adjuvant, neoadjuvant or metastatic. Therapies include surgery, radiation, and systemic treatments, such as chemotherapy and endocrine therapy [6,7,8]. New therapeutic options have been introduced and included in international therapeutic guidelines for BC treatment, including, for example, monoclonal antibodies that target human epidermal growth factor receptor 2 (HER2). Therapies that target nuclear receptors (NRs), such as the estrogen receptor (ER) and the progesterone receptor (PR), are very promising treatment options and have been shown to improve prognosis in studies conducted over many decades. Endocrine therapy regimens resulted in an approximately 30% decrease in BC-associated mortality, making them essential for the treatment of hormone receptor-positive (HR+) BC [9,10,11]. Moreover, clinical studies indicate a strong correlation between the expression of “classical steroid hormone receptors”, such as ER and PR, and disease progression [12,13,14,15,16]. Nevertheless, some tumors are resistant to these established therapeutic options, making the identification of new therapeutic targets central to our current research interests [17].

Currently, personalized BC therapy already includes NR-specific targeted therapies for both prevention and treatment [18]. NRs are activated via binding to amphiphilic hormones and function mainly as transcription factors in the nucleus [19,20]. Recent literature and data from our research group show that, in addition to the well-known NR, nuclear type II receptors, including retinoid X receptor alpha (RXRα), thyroid hormone receptors (TRs) and vitamin D receptor (VDR), play an important role in the pathophysiology of both BC and other cancers [21,22,23]. Studies of the role of NR in different intracellular compartments have shown that its specific prognostic value depends on subcellular localization [24]. Czogalla et al., demonstrated a direct association between cytoplasmic localization of VDR and poorer overall survival (OS) in ovarian cancer [25]. In the case of TRalpha, strong nuclear localization was reported to be a positive predictor of survival in epithelial ovarian cancer [22]. In addition, TRβ and TRβ1 were negative prognosticators if expressed in the cytoplasm [26]. In contrast, nuclear TRβ1 has been identified to have cancer-promoting activities in BC development [24]. Very recently, we found that cytoplasmic colocalization of RXRα and PPARγ, as well as cytoplasmic RXRα itself, are independent negative prognosticators in breast cancer patients [27,28].

Retinoids derived from vitamin A and co-activator molecules bind and activate RXRα, which then regulate the transcriptional activity of heterodimers with other nuclear receptors, like TR and VDR, and are activated in the nucleus to eventually promote its transcriptional activity after hormone binding [29]. In addition, thyroid hormones bind to its receptor monomer, and the RXRα/TR heterodimer acts as transcription factor [30,31,32,33]. As we recently discovered that cytoplasmic TRβ1 was correlated with favorable survival, whereas nuclear TRβ1 had a statistically significant correlation with poor outcome, we were interested in finding subcellular-specific analyses of TRα-expression in this study [24].

Due to its alleged contradictory role in BC prognosis, it appeared necessary to further investigate the behavior of TRα in BC. As cytoplasmic shuttling of type II nuclear receptors was found in many cases in our breast cancer collection, we focused on the relationship between TR-shuttling and survival. In addition, nucleocytoplasmic shuttling of TRα is a long-known phenomenon, albeit a phenomenon about which we lack understanding of its clinical relevance [34]. Former studies of our group showed that shuttling of nuclear type II receptors from the nucleus to the cytoplasm is accompanied by unfavorable outcomes in breast [27] and ovarian cancer [22]. Although thyroid hormone receptors were also analyzed in these tumor entities [22,24], to date, no study has identified TRα subcellular localization as a prognostic factor in human breast cancer samples. New findings may be promising in regard to individualized targeted BC therapy. In this study, we define the prognostic role of TRα and its isoforms α1 and α2 in association with cytoplasmic and nuclear expression of RXRα, respectively, in BC and relate the results to clinico-pathological criteria.

## 2. Materials and Methods

### 2.1. Patient Collective

The cohort used in this study included 272 primary BC tissues fixed in formalin and paraffin that were collected from patients operated on in the period 2000–2002 at the Department of Gynecology and Obstetrics, Ludwig Maximilian University of Munich, Germany.

Out of a total of 272 patients (Table 1), analyses of immuno-histochemical staining could be obtained in 249 cases due to the floated nature of tissue. After a follow-up period of up to 13 years, DFS, 10-year survival, and OS were statistically analyzed, and these follow-up data were extracted from the Munich Cancer Registry. Overall survival (OS) was defined as the time from randomization (date of surgery) to death. All patients who were not followed up or still alive at the time of assessment were censored. Disease-free survival (DFS) was defined as the time from randomization (date of surgery) to evidence of disease recurrence. Ten-year survival: the ten-year survival at randomization (time of surgery) was defined as the proportion of people who were still alive ten years after surgery.

The TNM classification of the Union for International Cancer Control (UICC) was completed to estimate primary tumor size (pT) [35,36], lymph node involvement (pN), and distant metastasis (pM). An experienced pathologist from the Department of Pathology at LMU determined the tumor’s grade and histological status. A tumor’s grade was determined according to the Bloom and Richardson grading system [37]. Hormone receptor status was determined through immuno-histochemistry on paraffin-embedded material. Cells were considered positive for hormone receptors when staining was positive in ≥10% of tumor cell nuclei. The Remmele and Stegner immunoreactive scoring system (IRS) was used [38]. 

### 2.2. Patient Treatment

As described previously [39,40], the main surgical treatment was breast conservation or modified radical mastectomy. Routine axillary dissections were performed on level I and II lymph nodes, while level III lymph nodes were only removed in cases with macroscopic metastatic lesions from the lower levels. For the diagnosis of lymph node metastases, individual embedded lymph nodes were examined in up to three levels. 

According to the guidelines of the Munich Cancer Treatment Center, patients in this study received chemotherapy in cases of lymph node involvement. Post-menopausal hormone receptor-positive patients received adjuvant endocrine therapy with tamoxifen (20 mg–30 mg/day). Pre-menopausal women received GnRH analogues during the later years of the follow-up period. Aromatase inhibitors were also used.

However, guidelines for surgery, radiotherapy, and chemotherapy changed significantly during the study observation period. Therefore, the authors did not provide details on cancer treatment.

### 2.3. Immuno-Histochemistry

According to the previously published and well-described methods [41,42,43], immuno-histochemistry of TRα, TRα1, and TRα2 was performed on formalin-fixed paraffin embedded sections. Specifically, a combination of pressure stove heating and a standard streptavidin–biotin–peroxidase complex with mouse/rabbit IgG Vectastain Elite ABC kit (Vector Laboratories, Burlingame, CA, USA) was used. The staining procedure was performed using commercially available mono- and poly-clonal antibody kits to detect TRα expression, as well as TRα1 and TRα2 (Table 2).

Paraffin-embedded tissue sections were, therefore, dewaxed in xylene for 15 min and rehydrated twice for 15 min in a solution containing 100% alcohol. Endogenous peroxidase activity was quenched via immersion in 3% hydrogen peroxide (H_2_O_2_) (Merck; Darmstadt, Germany) in methanol for 20 min. Once again, the sections were placed in a solution of 96% and 70% alcohol. After washing in phosphate-buffered saline (PBS), the sections were exposed for 10 min in a pressure cooker with sodium citrate buffer pH 6.0 to extract epitopes. To create a pH of 6.0, 0.1 M citric acid was diluted in 1 L of distilled water (solution A), and 0.1 M sodium citrate was diluted in 1 L of distilled water (solution B). The solution used contained 18 mL of solution A and 82 mL of solution B diluted with 900 mL of distilled water. This step was followed by washing the sections in distilled water and PBS. To prevent non-specific binding of primary antibodies (Table 2), sections were incubated with diluted normal serum (10 mL PBS that contained 150 μL horse serum, Vector Laboratories). The tissue sections were then incubated with the primary antibodies diluted in PBS (1:1000) for 1 h at room temperature. Sections were washed twice for 2 min in PBS. The sections were then incubated with the secondary antibody that bound the streptavidin–biotin–peroxidase complex (ABC complex) diluted in 10 mL PBS for 30 min, followed by multiple steps of washing with PBS and incubation with the ABC complex. Substrate staining was achieved using chromogenic 3,3′-diaminobenzidine (DAB; Dako, Glostrup, Denmark) for 1 min. After washing in PBS, sections were stained with Mayer’s acid hematoxylin for 2 min. Finally, the sections were rehydrated in increasing series of alcohol and coated with Eukit.

Serving as negative controls were human struma tissue sections incubated with pre-immune IgGs (supersensitive rabbit negative control, BioGenex, Fremont, CA, USA), which were used instead of the primary antibody (Figure 1a). As positive controls, we used struma (Figure 1b, TRα). Pictures were taken with a digital Charged Coupled Device (CCD) camera system (JVC, Tokyo, Japan). Additional control staining’s are shown in Appendix A.

### 2.4. Staining Evaluation (Immunoreactive Score)

To quantify the specific TRα, TRα1, and TRα2 immunoreactivity in the nuclei and cytoplasm, which corresponded to the distribution and intensity patterns, the well-established semi-quantitative immunoreactive scoring system (IRS) devised by Remmele and Stegner (IRS) [38] was used. Two independent blinded observers assessed the intensity and distribution of the staining response. In five cases (*n* = 1.8%), the judgment of the two independent observers differed. Both observers reassessed these cases together and ultimately interpreted the same result. Agreement before reassessment was reported as being 98.2%. The estimation method has been described previously and was used in several prior studies by our research group [41,42,43,44]. A Leitz light microscope (Immuno-histochemistry Type 307–148.001 512 686) (Wetzlar, Germany) and a 3CCD color camera (JVC, Victor company of Japan, Higashi-Osaka City, Japan) were used for staining analysis.

The IRS scoring system ranged from 0 to 12 points. To obtain an IR score, the staining intensity (score 0 = no staining, score 1 = weak staining, score 2 = moderate staining, score 3 = strong staining) and percentage of positively stained cells (0: no staining, 1: ≤10% of cells, 2: 11–50% of cells, 3: 51–80% of cells and 4: ≥81% of cells) were multiplied.

Nuclear and cytoplasmic TRα, TRα1, TRα2 staining were assessed in parallel, and nuclear and cytoplasmic IRS were determined separately. The endpoints for IRS were determined as follows: tissue samples that had an IRS of greater than 0 for nuclear or cytoplasmic expression of TRα, TRα1, and TRα2 were considered positive. An example of TRα2 is shown in Figure 2.

### 2.5. Ethical Approval

Tissue samples used in this study comprised material leftover after diagnosis was completed and sourced from the Archives of Gynecology and Obstetrics, Ludwig Maximilians University of Munich, Germany. All patients consented to participate in this study. All patient data and clinical information sourced from the Munich Cancer Registry were fully anonymized and coded for statistical analysis. The study was conducted in accordance with the standards of the 1975 Declaration of Helsinki. This study was approved by the Ethics Committee of the Ludwig Maximilians University of Munich, Germany (approval number 048-08). The authors were blinded to clinical information during the experimental analysis.

### 2.6. Statistical Analysis

Statistical analysis was performed using IBM Statistical Package for the Social Sciences (IBM SPSS Statistic v26.0 Inc., Chicago, IL, USA). Collected results were inserted into the SPSS database in an implicit manner and constructed a TC. The chi-square test was used to assess the distribution of clinico-pathologic variables. Correlations between immuno-histochemical staining results were determined via Spearman’s analysis. The non-parametric Kruskal–Wallis test was used to test for differences in cytoplasmic and nuclear expression of TRα, TRα1, and TRα2 in respect to the assigned prognostic markers. Life expectancy (in years), 10-year survival (in years), and disease-free survival (DFS) (in years) were compared using the Kaplan–Meier plot, and differences in patient survival times were tested for significance using the chi-square log-rank test statistic. The Cox regression model for survival was used for multivariate analysis, and the following factors were included: age at surgery, histology type, pT and pN from the TNM staging system, grading, and estrogen and progesterone receptor. Each parameter considered to be significant was indicated as *p* < 0.05. The *p*-value and the number of patients analyzed in each group were indicated for each graph.

## 3. Results

### 3.1. Correlation Analyses of TRα and TRα2 Staining for Breast Cancer Subtypes

Cytoplasmatic expression of TRα showed a significant correlation with Ki67 (Correlation coefficient (CC) = 0.158, *p* = 0.025) and the Luminal subtype of breast cancer (CC = 0.156, *p* = 0.027). Nuclear staining of TRα2 showed a significant negative correlation with the triple-negative subtype (CC = −0.266, *p* < 0.001) and a negative correlation with the basal and Her2 (luminal and non-luminal) subtypes (CC = −0.190, *p* = 0.002). In addition, cytoplasmic TRα and TRα2 showed a positive correlation with each other (CC = 0.168, *p* = 0.007). 

### 3.2. Cytoplasmic TRα Expression Is an Independent Negative Prognosticator for Overall Survival

Distribution of TRα in the cytoplasm of breast cancer is associated with significantly reduced overall (Figure 3a, *p* = 0.034) and 10-year survival (Figure 3b, *p* = 0.009), whereas the DFS shows no significant differences (Figure 3c, *p* = 0.522). Median FUP for DFS is 9.410 years for patients without TRα expression (CI 7.271–11.549) and 8.630 years for patients with TRα expression (CI 7.321–11.499).

Multivariate Cox regression identified cytoplasmic TRα as an independent negative prognostic factor influencing OS (HR 2.846, 95%CI 1.287–6.291, *p* = 0.010) (Table 3). For DFS, TRα showed a strong trend as an independent factor for recurrence (Table 4; *p* = 0.058).

Nuclear expression of TRα was not linked to significant survival changes (see Appendix A).

### 3.3. Nuclear TRα2 Expression Is Linked with Good Prognosis in Breast Cancer as Independent Prognosticator 

Nuclear TRα2 expression in BC tissue samples is associated with improved OS, 10-year survival, and DFS. The Kaplan–Meier curve visualized a positive association between OS, 10-year survival, and DFS (Figure 4) when expressing nuclear TRα2. The log-rank test calculated a *p* value of 0.029 for the OS, a *p* value of 0.014 for 10-year survival, and a *p* value of 0.043 for DFS. Median FUP for DFS is 8.070 years for patients without TRα2 expression (CI 4.475–11.665) and 10.850 years for patients with TRα2 expression (CI 7.788–13.912). Finally, multivariate Cox regression identified age at surgery and pN as independent survival factors (Table 5). For DFS (Table 6), no independent factor of DFS could be identified. 

Cytoplasmic expression of TRα2 was not correlated with different OS, 10-year survival, or DFS (all data are shown in Appendix A).

### 3.4. Cytoplasmic and Nuclear TRα1—Not for OS, 10-Year Survival and DFS

Nuclear and cytoplasmic TRα1 expression in BC tissue samples was not associated with impaired OS 10-year survival and DFS (all data shown in Appendix A). 

### 3.5. Survival Analyses for Nuclear TRα2 in Correlation to Specific Breast Cancer Subtypes 

As shown in the correlation analyses (Section 3.1) between TRαs and breast cancer subtypes, significant interactions exist. Therefore, we re-analyzed the TRα-survival rated corresponding to each subtype. We found that the protective effect of nuclear TRα2 for survival is only significant in the group of patients with Ki67 expression greater than 14% (Figure 5a). In addition, we found a significant positive effect of nuclear TRα2 expression on disease-free survival (DFS) in the Luminal A group (Figure 5b).

## 4. Discussion

The aim of this study was to evaluate the prognostic impact of subcellular expression of thyroid hormone receptors TRα, TRα1, and TRα2 determined in a large group of BC tissues and correlate the results with clinicopathologic criteria. So far, the role of thyroid hormones and their receptors (TR) in BC patients has not been sufficiently investigated [41].

TRs, which are members of the nuclear receptor superfamily, mediate the classical genomic actions of TH signaling in many tissues and regulate important developmental and homeostatic processes [24,45,46]. The TRα isoforms (TRα1 and TRα2) arise due to alternative splicing of the THRA gene [47]. TRα1 can bind thyroid hormone and mediate its biological effects [47,48]. TRα2 has no binding site for the thyroid hormone [47,49,50,51]. Unbound TRα2 is a weak antagonist of thyroid hormone-mediated transcription [47]. TRα expression was significantly associated with DFS in patients with breast cancer [52]. The expression of TRα2 correlated positively with the expression of ER and PR and correlated negatively with HER2 expression [47]. Low TRα2 expression was associated with inferior 5-year OS compared to high expression [47]. TRs heterodimerize with the retinoid X receptor (RXR) and act as ligand-dependent transcription factors [24]. TH activity is influenced by TR mutations, interactions with heterodimerization partners and coregulators, and expression of TR subtypes and their intracellular localization [53,54]. The shuttling of several TR isoforms between the nucleus and cytoplasm occurs, which may lead to specific TH-signaling activities in the nucleus, cytoplasm, or mitochondria [24,45,46]. Our previous studies showed that TRα and TRβ are expressed in the nuclei of breast cancer cells [41]. TRα2 was significantly associated with prognostic histo-pathological parameters, such as tumor size, axillary lymph node involvement, and grading and hormone receptor status [41]. There is a trend of TRα2 acting as an independent predictor of disease-free and overall survival (OS) [41]. In BRCA1-associated breast cancer, TRβ is a positive prognostic factor of OS at 5 years post-treatment, while TRα positivity predicts a reduced OS at 5 years posy-treatment [43]. Nuclear and cytoplasmic TRβ1 appear to be independent markers of either poor or good prognosis [24].

This paper represents the first study used to determine the respective prognostic roles of cytoplasmic and nuclear TRα expression in BC using a relatively large group of patients who received no treatment prior to surgery and completed long-term follow-up. The results of this study show that cytoplasmic TRα expression is a significant negative prognostic marker, while nuclear TR2α expression appears to be a protective factor.

To better understand the prognostic function of TRα in the pathogenesis of BC, this study focused separately on nuclear and cytoplasmic TRα, TRα1, and TRα2 expression in BC. Our study confirmed that TRα is expressed with a nuclear and cytoplasmic localization. Interestingly, nuclear and cytoplasmic forms of TRα may hereby exhibit opposite roles in mammary carcinogenesis.

Cytoplasmic expression of TRα in BC tissue was associated with significantly lower OS and ten-year survival rate, as well as tendential lower DFS, whereas nuclear expression of TRα revealed a tendential association with improved OS. In a multivariate analysis, cytoplasmic TRα is considered to be an independent negative prognostic factor of OS when adjusted to fit clinico-pathological parameters. TRα1 had no prognostic relevance, whereas nuclear TRα2 expression in BC tissue was associated with significantly longer OS, ten-year survival, and DFS. Additionally, a multivariate analysis identified nuclear TRα2 expression as an independent positive prognosticator of OS when adjusted to fit clinico-pathological parameters.

Interestingly, the results of our study confirm our former investigation into the subcellular localization of PPARγ [27,55] and RXRα [27,28] and its influence on survival in breast cancer patients. Within the latter studies, we showed that cytoplasmic localization of either PPARγ or RXRα is associated with shortened survival, whereas nuclear localization of both receptors leads to better outcomes. RXRα is, of course, also the heterodimeric partner of all thyroid hormone receptors [56,57,58]. The positive impact of TRα on the BC prognosis is possibly caused by heterodimerization with RXRα in the nucleus of breast cancer cells. Nuclear RXRα expression in breast cancer tissue leads to an improved OS, whereas cytoplasmic RXRα expression is significantly correlated with poor outcomes in terms of both OS and DFS [28]. The expression of cytoplasmic RXRα is correlated with more aggressive breast cancer types, whereas nuclear RXRα expression appears to be a protective factor [28]. Cytoplasmic RXRα also seems to be a negative prognosticator of Her-2neu-negative and triple-negative patients [28]. RXR- and PPARγ-forming heterodimers in breast cancer cells are reported to induce growth arrest and differentiation in breast cancer cells [29]. Depending on the localization of TRα and corresponding NR, specific responses, such as growth arrest and apoptosis, may be induced [59].

In contrast to the above-described situation, nuclear TRβ1 expression was related to poor outcomes, and cytoplasmic expression was related to favorable outcomes [24]. This finding is an exceptional result, because cytoplasmic expression of nuclear receptors is usually associated with reduced overall survival, and our investigation into the role of TRβ1 subcellular localization and outcomes in ovarian cancer showed that cytoplasmic TRβ1 is associated with poor outcomes [26]. Due to the fact that TRα2 has the strongest input in survival, while TRα1 has no impact on survival, the limited prognostic role of the general TRα antibody can be explained by the fact that it binds to both subtypes. This assumption is highly speculative because we have only limited information about the molecular role of both receptors.

The role of TRα2 in breast cancer was described previously by Sandsveden et al. [60], although no subcellular localization was analyzed. They stated that low tumor-specific TRα2 expression was, in their study, associated with prognostically unfavorable tumor characteristics and a higher mortality in breast cancer, though it was not independent of other prognostic factors [60].

In addition, in previous studies, we showed that TRα2 expression had a positive association with disease-free survival in multifocal breast cancer [42]. In that study, we did not investigate the subcellular localization. Furthermore, in an earlier study, our group found an inverse correlation between TRα2 and tumor size, lymph node involvement, histological grade, and hormone receptor expression, as well as a better disease-free survival rate among 82 women with higher levels of tumor-specific TRα2 [41]. Jerzak et al., also found evidence of an association between higher tumor-specific expression of TRα2 and favorable prognostic characteristics, as well as improved survival among 130 women with invasive breast cancer [47].

It is already known that TRα2 is an alternative splice product of the TRα primary transcript, whose unique carboxyl terminus does not bind thyroid hormones and, therefore, does not activate transcription [61]. In addition, the same group found that cellular localization studies demonstrated that phosphorylated TRα2 is primarily cytoplasmic, whereas unphosphorylated TRα2 is primarily nuclear. Since RNA binding is a property of unphosphorylated TRα2, the TRα2–RNA interaction likely represents a nuclear function of TRα2 [61]. Therefore, nuclear-expressed TRα2 that is associated with favorable outcome in breast cancer seems to be unphosphorylated. Cytoplasmic-expressed TRα2 does not have any prognostic value (see Appendix A) based on the results that we identified regarding the subcellular expression of TRα1.

Newer investigation showed that TRα1 acts as a new squamous-cell lung cancer diagnostic marker and poor prognosis predictor [62]. In addition, the TRα1 was the only receptor in a previous study of our group, showing a significant effect on unifocal BC. The Kaplan–Meier curve illustrated a worse DFS for unifocal BC patients when expressing the TRα1 [42]. In the whole cohort, cytoplasmic expression of TRα1 showed a trend of favorable survival (see Appendix A), albeit without reaching significance. The same observation can be defined for nuclear expression of TRα1.

The *THRA* gene encodes the TRα subtypes TRα1 and TRα2 [63,64]. In addition to antibodies detecting both TRα subtypes separately, there are also antibodies that detect TRα more generally [22]. In a former study, we found that TRα and its isoforms 1 and 2 were associated with different prognoses in ovarian cancer [22]. Nuclear TRα was associated with a reduced survival rate in clear-cell ovarian cancer, nuclear TRα1 was a positive prognosticator for all subtypes of ovarian cancer, nuclear TRα2 was a positive prognosticator for serous ovarian cancer, cytoplasmic TRα2 was associated with reduced OS in all subtypes, and cytoplasmic TRα1 was only associated with reduced OS in mucinous ovarian cancer [22].

Within this study, we showed that cytoplasmic-expressed TRα acts as a negative prognosticator for OS and 10-year survival in BC and is an independent negative prognosticator for OS, as analyzed via Cox-regression. Although nuclear TRα showed a trend of being a positive prognosticator in OS, as well as in 10-year survival, differences did not reach a level of significance (see Appendix A).

To gain further insight into potential individualized targeted treatment of BC, we assessed subcellular TRα expression in the context of clinico-pathological characteristics. Cytoplasmic TRα was significantly correlated with a worse prognosis in BC. Furthermore, nuclear TRα expression in BC tissue tended to be associated with a favorable prognosis, and nuclear expression TRα2 was a significant positive prognostic factor in BC. A more detailed investigation of intracellular localization of TRα and its isoforms 1 and 2 in BC, especially triple-negative breast cancer, that is characterized by worse OS and DFS and increased metastatic potential compared to other major BC subtypes might be of interest, because the identification of reliable predictive biomarkers is fundamental to finding new therapeutic strategies.

This study has some limitations related to its retrospective nature and the way in which TRα-isoforms were assessed. The immuno-histochemical study only allows a semiquantitative analysis. In addition, immunofluorescence techniques would allow a simultaneous investigation of all three TRα-isoforms in one cell. For that approach to take place, antibodies from different species are necessary. On the other hand, complicated immunofluorescence techniques are not easy to transfer to the daily routine pathology, given limited time, technical, and monetary possibilities.

Therefore, our data show that the TRα pathway could represent a promising therapeutic target in BC after additional investigations. The crosstalk between potential NR-ligands, as well as TRα and its isoforms TRα1 and TRα2, in relation to the therapeutic potential of BC should be investigated. Overall, these results demonstrate the complexity of the links between nuclear and cytoplasmic TRα expression and their impact on patient outcomes and emphasize the need for more detailed investigations into intracellular localization of TRα and its isoforms 1 and 2, as well as its interaction with other nuclear receptors in breast carcinoma, in order to understand its biomolecular function and role as a possible biomarker in BC diagnostics.

## 5. Conclusions

This study investigated the predictive value of nuclear localization of the TRα receptor and its isoforms TRα1 and TRα2, as opposed to its cytoplasmic expression, in human BC samples. Furthermore, we investigated the correlation between clinico-pathological criteria and patient outcomes and the subcellular localization of TRα and its isoforms. This paper represents the first retrospective cohort study used to determine the respective prognostic roles of cytoplasmic and nuclear TRα expression in sporadic breast cancer using a large clinical cohort of patients with long-term follow-up. TRα expression was found to play a contradictory role in BC prognosis depending on its intracellular localization: TRα expressed in the cytoplasm of BC tissues was negatively associated with prognostic factors, as well as patient survival, and was inversely related to the nuclear-localized TRα2.

In summary, nuclear receptors, such as TRα and its isoforms TRα1 and TRα2, seem to play roles in breast cancer oncogenesis. The importance of their subcellular expression and interaction with other members of the nuclear receptor family needs to be elucidated to find possible new target treatments for breast cancer in the future. Further investigations that study the biomolecular role of TRα in BC are ongoing within this study group.

## Figures and Tables

**Figure 1 cancers-15-03610-f001:**
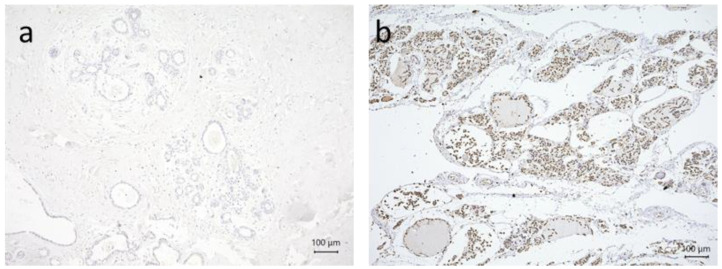
Immuno-histochemical staining serving as a negative (**a**) or positive control; (**b**) TRα struma. All pictures are 10× lens (scale bar = 100 µm) magnification.

**Figure 2 cancers-15-03610-f002:**
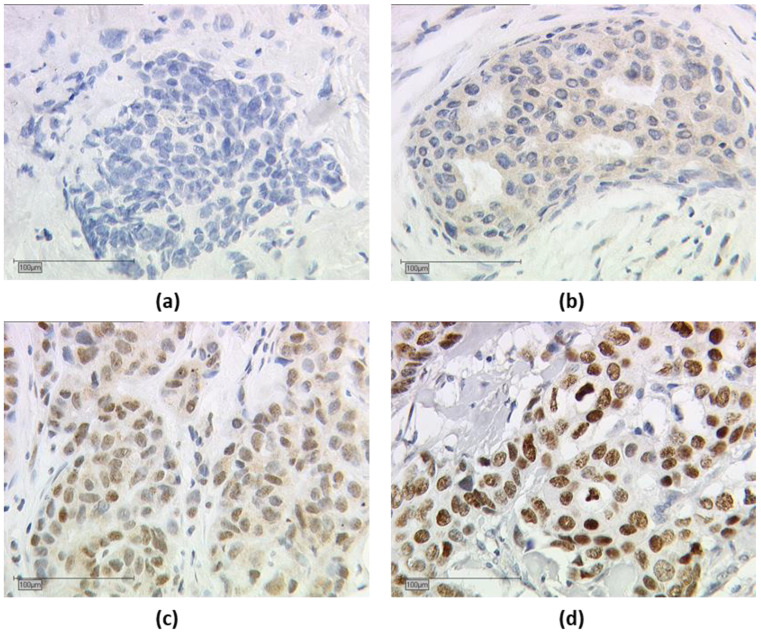
Immuno-histochemical staining of thyroid hormone receptor alpha 2 (TRα2). Immuno-histochemical staining of TRα2 in human breast cancer samples is illustrated in (**a**–**d**): (**a**) negative cytoplasmic and nuclear TRα2 expression, (**b**) positive cytoplasmic and negative nuclear TRα2 expression, (**c**) positive nuclear and cytoplasmic TRα2 expression, and (**d**) positive nuclear and negative cytoplasmic TRα2 expression; (**a**–**d**) shows a 25× (scale bar = 100 µm) magnification.

**Figure 3 cancers-15-03610-f003:**
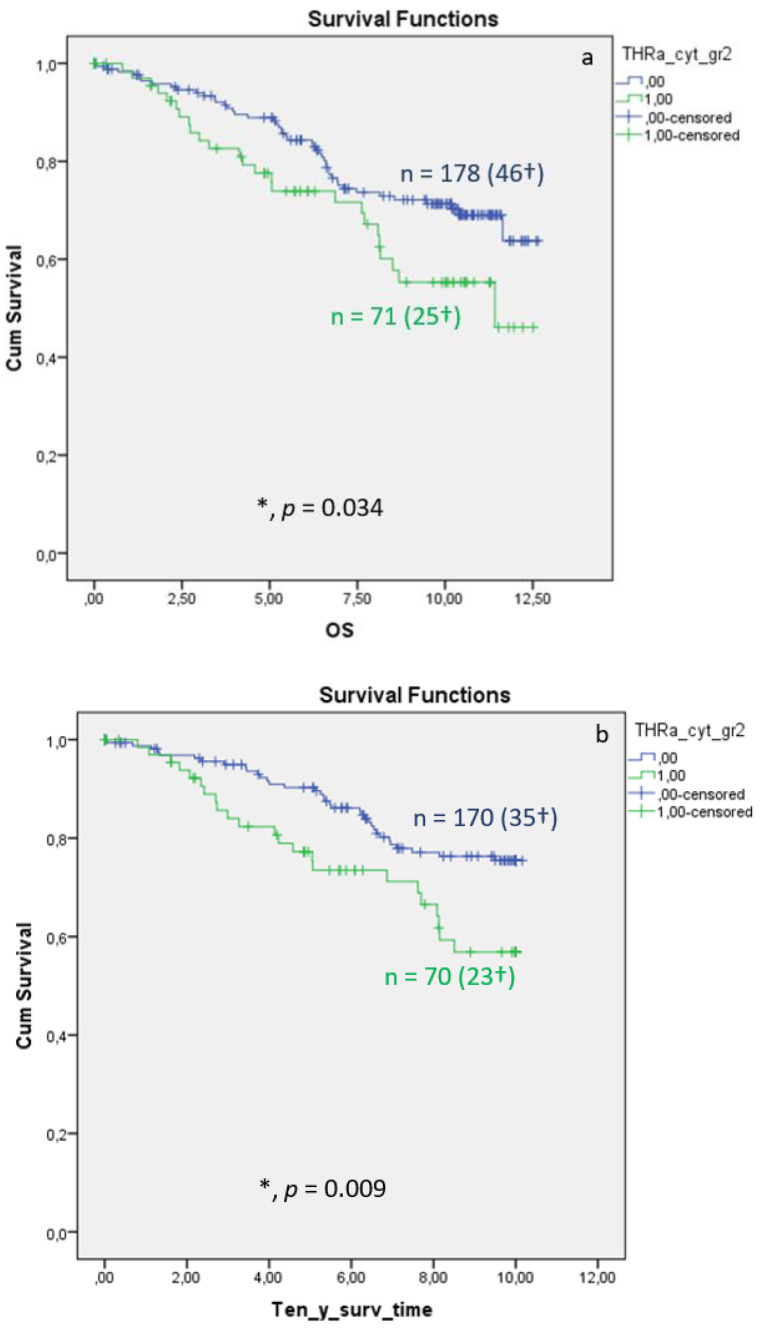
Kaplan–Meier survival analysis of cytoplasmic TRα expression in relation to overall survival (OS) (**a**), ten-year survival (**b**), and disease-free survival (DFS), with years shown in the X-axes and the cumulative survival rate (Cum Survival) shown in the Y-axes (**c**). n corresponds to the number of patients in each group, and the number in brackets (†) corresponds to the number of deceased patients or patients with a recurrence event for DFS in each group, asterisk (*) corresponds to significant differences (*p* < 0.05).

**Figure 4 cancers-15-03610-f004:**
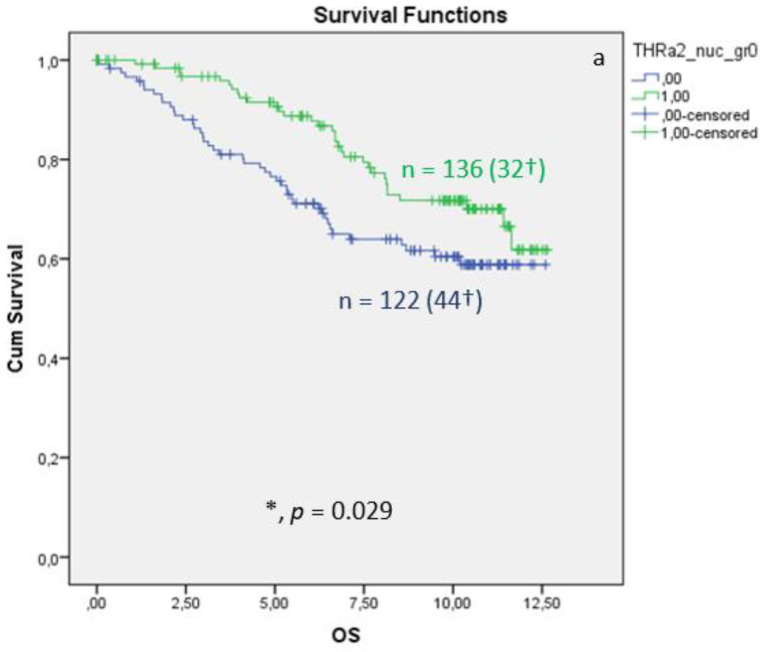
Kaplan–Meier survival analyses of nuclear TRα2 positive and negative expression in relation to overall survival (OS) (**a**), ten-year survival (**b**), and disease-free survival (DFS), shown as years in the X-axes and the cumulative survival rate (Cum Survival) in the Y-axes (**c**). n corresponds to the number of patients in each group, and the number in brackets (†) corresponds to the number of deceased patients or patients with a recurrence event for DFS in each group, asterisk (*) corresponds to significant differences (*p* < 0.05).

**Figure 5 cancers-15-03610-f005:**
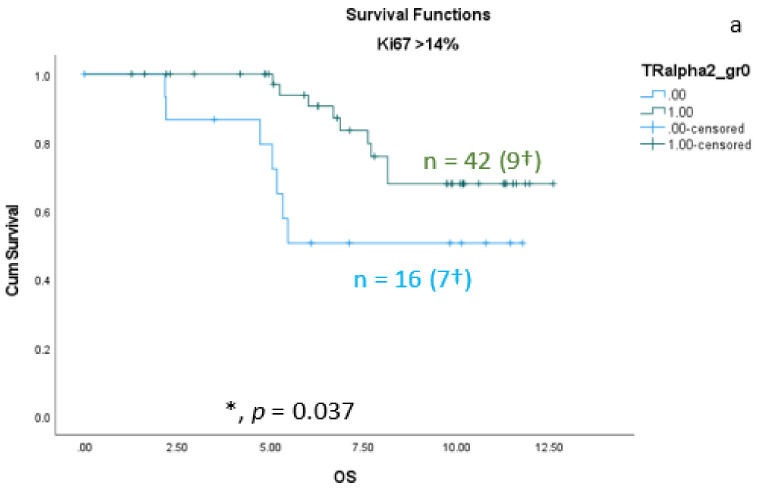
Kaplan–Meier survival analyses of nuclear TRα2-positive and TRα2-negative expression in relation to overall survival (OS) (**a**) in the group of breast cancer patients with Ki67 expression greater than 14% and disease-free survival (DFS) (**b**) in the group of Luminal A patients. Data are shown as years in the X-axes and the cumulative survival rate (Cum Survival) in the Y-axes. n corresponds to the number of patients in each group, and the number in brackets (†) corresponds to the number of deceased patients or patients with a recurrence event for DFS in each group, asterisk (*) corresponds to significant differences (*p* < 0.05).

**Table 1 cancers-15-03610-t001:** Clinical and pathological characteristics of all patients.

Clinical and Pathological Characteristics		%
Median Age (Years, *n* = 272)	57.00	Range 34.79–94.62
Median follow-up period (months, *n* = 272)	126	range 4–153
Histology c (*n* = 260)		
No Special Type (NST)	139	53.46%
NST with DCIS	74	28.46%
Other invasive	47	18.08%
ER status (*n* = 272)		
Positive	219	80.51%
Negative	53	19.49%
PR status (*n* = 272)		
Positive	160	58.82%
Negative	112	41.18%
HER2 status (*n* = 272)		
Positive	27	9.89%
Negative	246	90.11%
Molecular subtype (*n* = 272)		
Luminal A (Ki-67 ≤ 14%)	151	55.68%
Luminal B (Ki-67 > 14%)	60	21.98%
HER2-positive luminal	20	7.33%
HER2-positive non-luminal	7	2.56%
Triple negative	34	12.45%
Grade (*n* = 152)		
I	13	8.55%
II	95	62.50%
III	44	28.95%
Tumor size (*n* = 261)		
pT1	169	64.75%
pT2	78	29.89%
pT3	4	1.53%
pT4	10	3.83%
Lymph node metastasis (*n* = 256)		
Yes	112	43.75%
No	144	56.25%
Distant metastases d (*n* = 261)		
Yes	54	20.69%
No	207	79.31%
Local recurrence (*n* = 261)		
Yes	39	14.94%
No	222	85.06%

**Table 2 cancers-15-03610-t002:** Antibodies used in this study.

Antigen	Company	Antibody	Host	Catalog ID
TRα	Abcam	Polyclonal IgG	Rabbit	ab15543
TRα1	Abcam	Polyclonal IgG	Rabbit	ab53729
TRα2	Serotec	Monoclonal IgG1	Mouse	MCA2842

**Table 3 cancers-15-03610-t003:** Multivariate Cox regression analysis of cytoplasmic TRα expression regarding OS. Significant results are shown in bold (*p* < 0.05). HR: hazard ratio; CI: confidence interval.

OS	*p*-Value	Hazard Ratio [Exp(B)]	95% CI for Exp(B)	
			Lower	Upper
Age at surgery	0.272	1.017	0.987	1.049
Histological subtype	0.167	1.019	0.992	1.047
pT	0.456	1.104	0.851	1.432
pN	**0.011**	1.303	1.062	1.599
Grading	**0.027**	2.323	1.099	4.909
Estrogen receptor	**0.041**	0.423	0.186	0.965
Progesterone receptor	0.098	0.491	0.212	1.139
TRalpha	**0.010**	2.846	1.287	6.291

**Table 4 cancers-15-03610-t004:** Multivariate Cox regression analysis of cytoplasmic TRα expression regarding DFS. HR: hazard ratio; CI: confidence interval.

DFS	*p*-Value	Hazard Ratio [Exp(B)]	95% CI for Exp(B)	
			Lower	Upper
Age at surgery	0.425	0.988	0.960	1.018
Histological subtype	0.063	0.890	0.788	1.006
pT	0.214	1.176	0.910	1.520
pN	0.442	1.081	0.887	1.317
Grading	0.058	1.795	0.981	3.286
Estrogen receptor	0.433	0.727	0.327	1.614
Progesterone receptor	0.578	1.228	0.596	2.533
TRalpha	0.058	1.908	0.979	3.721

**Table 5 cancers-15-03610-t005:** Multivariate Cox regression analysis of nuclear TRα2 expression regarding OS. Significant results are shown in bold (*p* < 0.05); HR: hazard ratio; CI: confidence interval.

	*p*-Value	Hazard Ratio [Exp(B)]	95% CI for Exp(B)	
			Lower	Upper
Age at surgery	**0.031**	1.028	1.003	1.054
Histological subtype	0.719	1.005	0.980	1.030
pT	0.159	1.171	0.940	1.458
pN	**0.003**	1.311	1.098	1.565
Grading	0.337	1.344	0.735	2.458
Estrogen receptor	0.229	0.608	0.270	1.367
Progesterone receptor	0.298	0.667	0.311	1.430
TRalpha2	0.154	0.835	0.652	1.070

**Table 6 cancers-15-03610-t006:** Multivariate Cox regression analysis of nuclear TRα2 expression regarding DSF. HR: hazard ratio; CI: confidence interval.

DFS	*p*-Value	Hazard Ratio [Exp(B)]	95% CI for Exp(B)	
			Lower	Upper
Age at surgery	0.329	0.986	0.958	1.014
Histological subtype	0.110	0.909	0.808	1.022
pT	0.165	1.192	0.930	1.529
pN	0.205	1.129	0.936	1.363
Grading	0.061	1.684	0.976	2.906
Estrogen receptor	0.604	0.811	0.367	1.792
Progesterone receptor	0.547	1.237	0.619	2.470
TRalpha2	0.478	0.948	0.819	1.098

## Data Availability

Not applicable.

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
