# Peer review of "Cytoplasmic Localization of Thyroid Hormone Receptor (TR) Alpha and Nuclear Expression of Its Isoform TRα2 Determine Survival in Breast Cancer in Opposite Ways"

_cancers, 2023, doi:10.3390/cancers15143610_

Round 1

Reviewer 1 Report

The following have to be added to the manuscript:

1. time-to event end-points (OS, DFS, 10-yr survival) definitions; 

2. median FUP time with the corresponding CI;

3. distribution of the survival status.

Tables 3 and 5 have little value, mean is not an appropriate statistic for OS (DFS).  It looks like one of the tables in "Table 5" is displaying multivariable Cox regression results with similar selection of the plausible predictors as in Table 4? Why the multivariable Cox model was not utilized for all predictors used in tables 4 and 5? Where are the results of the multivariable regression model for the DFS? Figures and tables are confusing, all need major editing.

Please use standard terms in the paper, 'sign' and 'sig.' should be p-value; edit x and y axis of the graphs. Tables 3-5 should be redesigned to fit the standards of reporting OS and DFS. Graphs are too small to read. Adding number of patients as risk would enhance the information.

Author Response

Reviewer 1:

The following have to be added to the manuscript:

  1. time-to event end-points (OS, DFS, 10-yr survival) definitions; 

Answer: We added the definitions for OS, DFS and 10-year survival in Chapter 2.1:

Overall survival (OS) is defined as the time from randomization (date of surgery) to death. All patients who were not followed up or were still alive at the time of assessment were censored.

Disease-free survival (DFS) is defined as the time from randomization (date of surgery) to evidence of disease recurrence.

10-yr survival: The 10-year survival at randomization (time of surgery) is defined as the proportion of people who are still alive 10 years after surgery.

  1. median FUP time with the corresponding CI;

Answer: SPSS did evaluate median FUP time with the corresponding CI only for DFS for both TRa and TRa2. We added these data:

Median FUP for DFS is 9.410 years for patients without TRa expression (CI 7.271-11.549) and 8.630 years for patients with TRa expression (CI 7.321-11.499).

Median FUP for DFS is 8.070 years for patients without TRa2 expression (CI 4.475-11.665) and 10.850 years for patients with TRa2 expression (CI 7.788-13.912).

  1. distribution of the survival status.

Answer: We added the distribution of the survival status as numbers in the Kaplan-Meier graphs.

Tables 3 and 5 have little value, mean is not an appropriate statistic for OS (DFS).  It looks like one of the tables in "Table 5" is displaying multivariable Cox regression results with similar selection of the plausible predictors as in Table 4? Why the multivariable Cox model was not utilized for all predictors used in tables 4 and 5? Where are the results of the multivariable regression model for the DFS? Figures and tables are confusing, all need major editing.

Answer: We deleted former table 3 and 5 and kept the multivariate Cox-Regression analyses. We re-analyzed the Cox-regression in former table 5 as demanded with the same parameters as now shown in table 3 and 5. Finally, we included Cox-regression for DFS (table 4 and 6).

Comments on the Quality of English Language

Please use standard terms in the paper, 'sign' and 'sig.' should be p-value; edit x and y axis of the graphs. Tables 3-5 should be redesigned to fit the standards of reporting OS and DFS. Graphs are too small to read. Adding number of patients as risk would enhance the information.

Answer: We used standard terms as demanded. We enhanced the size of the graphs. X- and Y-axes in Kaplan-Meier analyses are standard of the SPSS program. We included an explanation in the Figure legends. We added the number of patients of each group in the Kaplan-Meier graph.

Reviewer 2 Report

The authors continue on a set of other studies on the role of selected nuclear receptors in breast cancer survival. The localization studies for TR isoforms is very interesting and relevant.

one unclear thing to this reviewer is why the authors did not stratify the results based on breast cancer subtype, as the patient population is well annotated.  this could be very relevant especially if a therapeutic approach has to be proposed. 

In terms of the IHC, are the antibodies validated to be that specific for the different isoforms? It would have been a lot more informative to perform double immunofluorescence instead as a direct comparison in a quantitative manner could be performed.  To me this is the biggest limitation of the study, as quantification of IHC with the method proposed is semi-quantitative at best and cannot be automated/utilized in an unbiased manner across centers. There also the need to add more information on how the imaging was performed (microscope, fields of view, magnification etc).

In figure 3 etc it is unclear how the groups were stratified (IRS score?) and how the decision was made for the threshold chosen.

Author Response

Reviewer 2

The authors continue on a set of other studies on the role of selected nuclear receptors in breast cancer survival. The localization studies for TR isoforms is very interesting and relevant.

Answer: Thank you very much for your evaluation.

one unclear thing to this reviewer is why the authors did not stratify the results based on breast cancer subtype, as the patient population is well annotated.  this could be very relevant especially if a therapeutic approach has to be proposed. 

Answer: We included a correlation analysis of both, the cytoplasmic TRa and the nuclear TRa2 staining on breast cancer subtype. We included the results in a Correlation subchapter in the Result part:

3.1. Correlation analyses of TRa and TRa2 staining with breast cancer subtype

Cytoplasmatic expression of TRa showed a significant correlation to Ki67 (Correlation coefficient (CC) = 0.158, p = 0.025) and to the Luminal subtype of breast cancer (CC = 0.156, p = 0.027). Nuclear staining of TRa2 showed a significant negative correlation to the triple negative subtype (CC = -0.266, p < 0.001) and a negative correlation to the basal and Her2 (luminal and non-luminal) subtype (CC = -0.190, p = 0.002).

Answer: In addition, we also included a subchapter on survival analyses of TRa2 in specific breast cancer subtypes:

3.5. Survival analyses for nuclear TRα2 in correlation to specific breast cancer subtypes

As shown in the correlation analyses (3.1.) between TRas and breast cancer subtypes, there exist significant interactions. Therefore, we re-analyzed TRa-survival rate corresponding to the subtype. We identified, that the protective effect of nuclear TRa2 for survival is significant only in the group of patients with Ki67 expression greater than 14% (Figure 5a). In addition, we also found a significant positive effect of nuclear TRa2 expression on disease free survival (DFS) in the Luminal A group (Figure 5b).

Figure 5. Kaplan-Meier survival analysis analyses nuclear TRα2 positive and negative expression in relation to overall survival (OS) (a) in the group of breast cancer patients with Ki67 expression greater than 14%. and disease-free survival (DFS) (b) in the group of Luminal A patients. Data are shown in years in the X-axes and the cumulative survival rate (Cum Survival) in the Y-axes. n corresponded to the number of patients in each group, number in brackets correspond to the number of deceased patients or patients with recurrence event for DFS in each group.

In terms of the IHC, are the antibodies validated to be that specific for the different isoforms?

Answer: We performed positive and negative control staining with all antibodies used for the TRalpha isoform analyses. Specific information on the tissue used (we used the human protein atlas for positive control tissue information) and pictures of the staining are included in the Suppl. Figures.

It would have been a lot more informative to perform double immunofluorescence instead as a direct comparison in a quantitative manner could be performed.  To me this is the biggest limitation of the study, as quantification of IHC with the method proposed is semi-quantitative at best and cannot be automated/utilized in an unbiased manner across centers.

Answer: Yes indeed, double or triple immunofluorescence analyses would enable us to identify all 3 isoforms in one cell. For that approach we need primary antibodies from 3 different species or at least 3 different IgG isoforms. We do not have these antibodies in our laboratories. Due to limitation of revision time, we will include a limitation paragraph in the Discussion part. The suggestion to perform such an analysis will be realized in one of our next studies that are planned on cell cultures. We explicitly thank the reviewer for this suggestion.

Limitation of our study:

“This study has some limitations, considering its retrospective nature, and the way TRa-isoforms were assessed. The immunohistochemical study allows only a semiquantitative analysis. In addition, immunofluorescence techniques would allow a simultaneous investigation of all three TRa-isoforms in one cell. For that approach antibodies from different species are necessary. On the other hand, complicated immunofluorescence techniques are not easy to transfer to the daily routine pathology with limited time, technical and monetary possibilities.”

There also the need to add more information on how the imaging was performed (microscope, fields of view, magnification etc).

Answer: This information was included in Material and Methods section:

To quantify the specific TRα, TRa1, TRa2 immunoreactivity in the nuclei and the cytoplasm, meaning the distribution and intensity patterns, a well-established semi-quantitative immune-reactive scoring system (IRS) of Remmele and Stegner (IRS) [1] was used. Two independent, blinded observers assessed the intensity and distribution of the staining response. In five cases (n = 1.8%), the judgment of the two independent observers differed. Both observers reassessed this case together and ultimately interpreted the same result. Agreement before reassessment was reported as 98.2%. The estimation method has been described previously and has been used in several studies by our research group. [2-5]. A Leitz light microscope (Immunohistochemistry Type 307-148.001 512 686) (Wetzlar, Germany) and a 3CCD color camera (JVC, Victor company of Japan, Japan) were used for staining analysis.

The IRS scoring system ranges from 0 to 12 points. To obtain the IR score, the staining intensity (score 0 = no staining, score 1 = weak staining, score 2 = moderate staining, score 3 = strong staining) and the percentage of positively stained cells (0: no staining, 1: ≤ 10% of cells, 2: 11-50% of cells, 3: 51-80% of cells and 4: ≥ 81% of cells) are multiplied. Nuclear and cytoplasmic TRα staining, TRa1, TRa2 was assessed in parallel and nuclear and cytoplasmic IRS were determined separately. The endpoints for IRS were determined as follows: Tissue samples that had an IRS greater than 0 for nuclear or cytoplasmic expression of TRα, TRa1, TRa2 was considered positive.

In figure 3 etc it is unclear how the groups were stratified (IRS score?) and how the decision was made for the threshold chosen.

Answer: To obtain the best cut-off value for the survival analysis in this case, we used ROC-Analyses as described in one of our former studies [6].

References

  1. Remmele, W., and H. E. Stegner. "[Recommendation for Uniform Definition of an Immunoreactive Score (Irs) for Immunohistochemical Estrogen Receptor Detection (Er-Ica) in Breast Cancer Tissue]." Pathologe 8, no. 3 (1987): 138-40.
  2. Ditsch, N., B. Toth, I. Himsl, M. Lenhard, R. Ochsenkuhn, K. Friese, D. Mayr, and U. Jeschke. "Thyroid Hormone Receptor (Tr)Alpha and Trbeta Expression in Breast Cancer." Histol Histopathol 28, no. 2 (2013): 227-37.
  3. Heublein, S., D. Mayr, A. Meindl, M. Angele, J. Gallwas, U. Jeschke, and N. Ditsch. "Thyroid Hormone Receptors Predict Prognosis in Brca1 Associated Breast Cancer in Opposing Ways." PLoS One 10, no. 6 (2015): e0127072.
  4. Ditsch, N., D. Mayr, M. Lenhard, C. Strauss, A. Vodermaier, J. Gallwas, D. Stoeckl, M. Graeser, T. Weissenbacher, K. Friese, and U. Jeschke. "Correlation of Thyroid Hormone, Retinoid X, Peroxisome Proliferator-Activated, Vitamin D and Oestrogen/Progesterone Receptors in Breast Carcinoma." Oncol Lett 4, no. 4 (2012): 665-71.
  5. Zehni, A. Z., F. Batz, A. Vattai, T. Kaltofen, S. Schrader, S. N. Jacob, J. N. Mumm, H. H. Heidegger, N. Ditsch, S. Mahner, U. Jeschke, and T. Vilsmaier. "The Prognostic Impact of Retinoid X Receptor and Thyroid Hormone Receptor Alpha in Unifocal Vs. Multifocal/Multicentric Breast Cancer." Int J Mol Sci 22, no. 2 (2021).
  6. Zhu, J., F. Trillsch, D. Mayr, C. Kuhn, M. Rahmeh, S. Hofmann, M. Vogel, S. Mahner, U. Jeschke, and V. von Schonfeldt. "Prostaglandin Receptor Ep3 Regulates Cell Proliferation and Migration with Impact on Survival of Endometrial Cancer Patients." Oncotarget 9, no. 1 (2018): 982-94.

Round 2

Reviewer 2 Report

we thank the authors for addressing all the concerns from the first round of revisions. nothing further is requested.